# Finite-Time Backstepping Control for Electro-Hydraulic Servo System via Extended State Observer with Perturbation Estimation Performance Improvement

**Fanliang Meng** [1] , **Hao Yan** [1,2,*], **Jiafeng Li** [1] **and Xu Liu** [1]

1   School of Mechanical, Electronic and Control Engineering, Beijing Jiaotong University, Beijing 100044, China
2   Key Laboratory of Vehicle Advanced Manufacturing, Measuring and Control Technology, Beijing Jiaotong University, Ministry of Education, Beijing 100044, China
*   Correspondence: hyan@bjtu.edu.cn

**Abstract:** The extended state observer (ESO) has been widely used in the state and perturbation estimation of the electro-hydraulic servo system. It was found that there was a controlled quantity in the transfer function between the perturbation estimation value and the disturbance. This indicates that the traditional linear ESO's estimation of the disturbance is affected by the change in the control input. To solve this problem, a new structure ESO for a hydraulic system (LHYESO) was designed by introducing the hydraulic system's load pressure and system model. The corresponding frequency domain analysis results show that it eliminates the control input in the transfer function and reduces the dependence of the high-frequency domain range of the perturbation estimation on the significant observer gain. To improve the estimation speed, a finite-time convergent ESO for hydraulic systems (FTHYESO) was proposed based on the structure of LHYESO, and it was proved that the observation error converged to a sufficiently small value during a finite time. Moreover, a finite-time backstepping controller has been designed by using the Lyapunov method to guarantee the rapidity and precise response of the hydraulic servo system. Finally, the experiment results show the effectiveness of the proposed method.

**Keywords:** extended state observer; electro-hydraulic servo system; perturbation estimation performance improvement; finite-time convergent ESO; finite-time backstepping controller



## 1. Introduction

Hydraulic servo drive plays an irreplaceable role in robotics [1], metallurgy [2], and equipment manufacturing [3] because of its high power density and high-frequency response [4]. However, hydraulic servo systems typically have uncertainties, such as unmodeled dynamics, external disturbances, and time-varying friction [5,6], which brings difficulties in improving the tracking performance. Hence, to deal with all these uncertainties, adaptive parameter control [7–9], H-infinity control [10], robust control [11,12], and sliding mode control [13–15] have been developed for hydraulic servo systems. These nonlinear control methods successfully solve the hydraulic servo systems' uncertainty and disturbance problems successfully in some conditions. Nevertheless, almost of all the above literature use full-state control schemes, which means that hard-to-measure system states such as velocity also need to be known. Unfortunately, this is difficult to meet in engineering practice due to structural and cost constraints.

In order to estimate the system states which are difficult to measure or cannot be measured, several observers have been developed [16–18]. In particular, the extended state observer (ESO) regards unknown parts and disturbance as an extended state and names its total perturbation [19–21]. ESO is proposed not only to estimate the unmeasured state of the system but also to observe the unknown parts and disturbances of the system [19–21].

Therefore, ESO has been widely used in the electro-hydraulic servo system to overcome complex total perturbation.

Considering the limitations of using sensors and in order to make the observer structure simple, the traditional extended state observer (TESO) only employs the two-system information of the control input and system output to estimate states, including total perturbation [22]. A TESO is constructed in [23] based on the hydraulic servo system's position signal and control input. In order to improve the observation accuracy, there are two main methods: the first is to use high-gain TESO or change the fixed gain to variable gain, and the second is to change the structure of the TESO. In [24], a high-gain extended state observer is proposed to accurately estimate the full state and disturbance of the hydraulic system. However, high-gain ESOs increased sensitivity to sensor noise while improving their observation performance. Furthermore, variable gain TESO is designed in [25] to reduce the negative influence of noise by constant large ESO gain parameters. An error-based observer gain adaptive law is proposed in [26,27]. When the observation error is large, the observer gain also increases to achieve fast estimation. For the flexible and intelligent adjustment of parameters, neural networks and TESO are combined in [28] to handle the problem that appropriate ESO parameters are difficult to obtain. On the other hand, model-assisted ESOs are designed in [29] based on the hydraulic system model to improve system state observation accuracy. In [30], two extended state observers are constructed to improve the tracking performance of hydraulic servo systems. A dual extended stated observer is explored to simultaneously estimate the matched and unmatched disturbances in the dynamic model [31]. Furthermore, an ideal ESO is constructed in [32] by using a hydraulic servo system's position, velocity, acceleration, and control input, and its effectiveness is verified in simulation. However, in engineering practice, the velocity and acceleration of the hydraulic servo systems are usually difficult to obtain or require high economic costs, which bring difficulties to their practical applications. Additionally, most of the extended state observers with these new structures are asymptotically stable, which means that the observation error may converge to the neighborhood of zero for a long time, which may also affect the accuracy of the feedback control.

The method of improving the observation accuracy by adjusting the parameters of TESO has achieved good research results, but in terms of the structural design of the ESO, the research on the observation accuracy of the hydraulic system by the TESO is not enough. In [33], the transfer function of the total perturbation estimation error shows that the control input affects the perturbation observation error. This means that changes in the control input will affect the observation accuracy of the total perturbation, which in turn affects the accuracy of the state feedback control. Regarding how to improve the observation accuracy, [32] gives us an idea. Perturbation can be estimated more accurately if the hydraulic system's information is available. The load pressure of the hydraulic system is easy to measure, and part of the model information can also be known. It is worth studying to combine these two parts with the design of the new structure of ESO. Moreover, based on the analysis of the new structure ESO, in order to improve the convergence speed, the design method of finite time convergence is worthy of further exploration.

The backstepping technique is an effective approach widely used in the controller design for complex hydraulic systems. A backstepping controller with good robustness is designed in [34] to control an electro-hydraulic brake system with highly nonlinear dynamic characteristics, and the simulation results show good characteristics. Based on the backstepping framework, a nonlinear controller is designed in [35] for an electro-hydraulic rotary actuator using the Lyapunov theory. In [36], a backstepping controller is designed to make the multi-joint hydraulic manipulator track desired path points. An adaptive robust impedance controller based on a traditional extended state observer and backstepping method is designed in [37] for the hydraulic mining hydraulic excavator boom to save energy and ensure a smooth digging process. To surmount the problem of unmodeled disturbances and a valve dead zone, the system model and a backstepping technique were employed in [38] to assure the ultimately bounded tracking performance. To solve the

problem of the valve deadband and output constraint in the hydraulic exoskeleton control system, a research method of a nonlinear backstepping control strategy is proposed in [39]. These simulations and experiments show that the backstepping approach can effectively resist hydraulic system disturbances and provide accurate tracking performances. However, these main research results have used the Lyapunov function method to design the virtual control input and the final real control input but have not discussed how to design the control input to make the hydraulic system converge in a finite time.

This paper proposes a finite-time backstepping control law based on an ESO with an improved observer performance. First, a new-structure linear ESO for a hydraulic system (LHYESO) is designed by introducing the load pressure of the system state and the system model. Then, to improve the estimation speed of the states and disturbance for the hydraulic systems, a finite time convergent ESO for hydraulic systems (FTHYESO) is proposed based on the structure of LHYESO, and the proof process of finite-time convergence is also given in detail. Moreover, a finite time backstepping controller has been designed by the Lyapunov method to guarantee the rapidity and precise response of the hydraulic servo system based on FTHYESO. Finally, the experiment results are given to illustrate the effectiveness of the proposed method.

## 2. Research Methods

### 2.1. System Model

In this paper, the schematic diagram of the hydraulic servo system is illustrated in Figure 1. As shown, a servo valve-controlled double-rod hydraulic actuator overcomes the elastic force, damping force, and external load force to drive a constant inertia load. In this hydraulic servo system, the supply oil pressure $P_s$ is stable and constant with the guarantee of the relief valve and accumulator. The return pressure $P_r$ is usually small compared to the supply pressure and can be ignored. Position and pressure sensors are, respectively, employed to produce the piston position and measure the two internal chambers of the actuator. The control objective is to conquer disturbance and to have the mass load track any smooth desired trajectory as accurately as possible, with the real-time oil pressure and actuator position measurement.

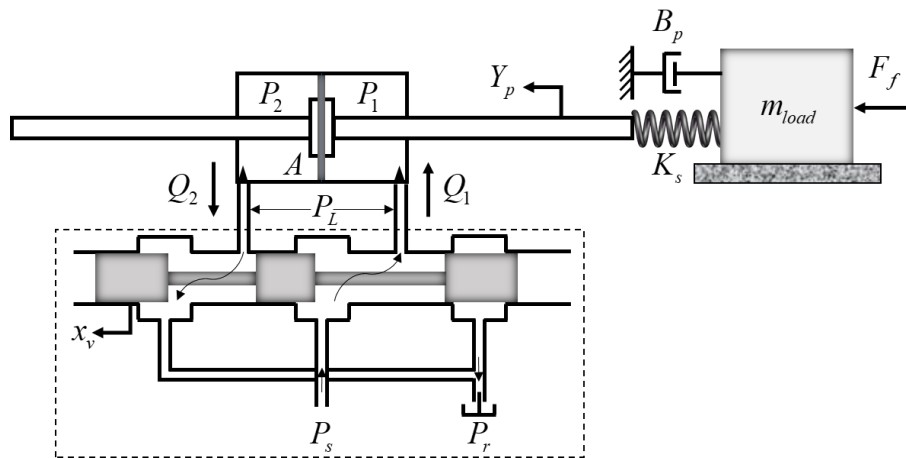

**Figure 1.** The architecture of the considered hydraulic servo system.

Considering [40], the dynamics of the mass load driven by a double-rod hydraulic actuator can be established as:

$$(P_1 - P_2)A = m_{load}\ddot{Y}_p + B_p\dot{Y}_p + K_sY_p - F_f \tag{1}$$

where $P_1$ and $P_2$ are the two chambers' pressure of the hydraulic actuator, $P_1 - P_2$ can be represented by load pressure $P_L$; $A$ is the effective area of the hydraulic actuator piston; $m_{load}$ and $Y_p$ represent the inertia and displacement of the mass block, respectively; $B_p$ is the

viscous damping coefficient of the system; $K_s$ is the elastic load stiffness; and $F_f$ indicates the disturbances caused by unconsidered external disturbance and friction force.

According to Dalla and Wang [40,41], the pressure dynamics can be written as follows, considering the compressibility and leakage of the oil:

$$Q_L - \frac{V_t}{4\beta_e}\dot{P}_L - C_{tp}P_L = A\dot{Y}_p \tag{2}$$

where $Q_L$ represents the load flow; $V_t$ is the total control volumes of the actuator chambers; $\beta_e$ is the effective oil bulk modulus; and $C_{tp}$ is the total linear leakage coefficient of the actuator related to the load pressure.

The high-response servo valve is considered here; the response time of it can be ignored, and the control input current $u$ is added to the servo valve as proportional to the spool position $x_v$, i.e., $x_v = K_{sv}u$ with a positive coefficient $K_{sv}$ [41]. Hence, the load flow can be represented as:

$$Q_L = K_q K_{sv} u - K_c P_L \tag{3}$$

where $K_q$ is the flow gain of the servo valve; $K_c$ is the flow-pressure coefficient.

When the response frequency of the servo valve is much higher than the motion frequency, the equations for pressure and flow can be linearized [40,41]. The response frequency of the servo valve selected in this experiment is 100 Hz, which is much higher than the actual experiment frequency, so this paper linearizes Equations (1)–(3).

Denote the state as $x = [x_1, x_2, x_3]^T \triangleq \left[Y_p, \dot{Y}_p, P_L\right]$, then the whole hydraulic servo system can be rewritten in a state-space form as:

$$\begin{aligned}
\dot{x}_1 &= x_2 \\
\dot{x}_2 &= \frac{1}{m_{load}}(Ax_3 - B_p x_2 - K_s x_1) \\
\dot{x}_3 &= \frac{4\beta_e}{V_t}(K_q K_{sv} u - Ax_2 - K_{ce}x_3 + \frac{K_{ce}}{A}F_f + \frac{V_t}{4\beta_e A}\dot{F}_f) + \Delta
\end{aligned} \tag{4}$$

where $K_{ce} = K_c + C_{tp}$ represents the total flow-pressure coefficient; $\Delta$ is the unmodeled part of hydraulic servo systems.

An extended state variable $x_4 \triangleq F_{Tf} = 4\beta_e K_{ce}F_f/V_t A + \dot{F}_f/A + \Delta$ is set for the hydraulic servo system (4) and satisfying $\dot{x}_4 = \dot{F}_{Tf}$. $F_{Tf}$ is the unknown total perturbation of the hydraulic servo system and is continuous and bounded, i.e., $\left|\dot{F}_{Tf}\right| < F_d$. The system (4) becomes a fourth-order system. The system with an extended state is rewritten as:

$$\begin{aligned}
\dot{x}_1 &= x_2 \\
\dot{x}_2 &= -l_1 x_1 - l_2 x_2 + l_3 x_3 \\
\dot{x}_3 &= -s_2 x_2 - s_3 x_3 + x_4 + b_0 u \\
\dot{x}_4 &= \dot{F}_{Tf}
\end{aligned} \tag{5}$$

where $l_1 = K_s/m_{load}$, $l_2 = B_p/m_{load}$, $l_3 = A/m_{load}$; $s_2 = 4\beta_e A/V_t$, $s_3 = 4\beta_e K_{ce}/V_t$, $b_0 = 4\beta_e K_{sv}K_q/V_t$.

System (5) is written as the following matrix equation:

$$\dot{x} = Ax + Bu + B_1\dot{F}_{Tf} \tag{6}$$

where $x = [x_1, x_2, x_3, x_4]^T$, $B = [0, 0, b_0, 0]^T$, $B_1 = [0, 0, 0, 1]^T$.

### 2.2. Problem of Traditional Extended State Observer for Hydraulic Servo System

To obtain the estimation $F_{Tf}$, which is used to design various control strategies, according to Han [19], the ordinarily used linear fourth-order ESO is written as:

$$\dot{z} = A_t z + Bu + Bet Y_p \tag{7}$$

where $z = [z_1, z_2, z_3, z_4]^T$; $A_t = \begin{bmatrix} -\beta_1 & 1 & 0 & 0 \\ -\beta_2 & 0 & 1 & 0 \\ -\beta_3 & 0 & 0 & 1 \\ -\beta_4 & 0 & 0 & 0 \end{bmatrix}$; $Bet = [\beta_1, \beta_2, \beta_3, \beta_4]^T$ are coefficientsde-

signed to make ESO Hurwitz stable.

The most critical task of the ESO is to make $z_4$ track $F_{Tf}$, quickly and accurately. Then, we derived the transfer function of the total perturbation and its observation. The laplace transform for the hydraulic servo system state is represented by Equation (6):

$$Y_p(s) = x_1(s) = [1,0,0,0](sI - A)^{-1}[Bu(s) + B_1 s F_{Tf}(s)]$$
$$P_L(s) = x_3(s) = [0,0,1,0](sI - A)^{-1}[Bu(s) + B_1 s F_{Tf}(s)] \tag{8}$$

Replacing $Y_p(s)$ in $z_4$ of Formula (7) with $Y_p(s)$ in Formula (8), we obtained the relationship between the total perturbation and its estimated value as follows:

$$z_4(s) = \frac{(a_0 F_{Tf}(s) - b_1 u(s))s^3 - b_2 u(s)s^2 - b_3 u(s)s - b_4 u(s)}{s^7 + c_1 s^6 + c_2 s^5 + c_3 s^4 + c_4 s^3 + c_5 s^2 + c_6 s + c_7} \tag{9}$$

where $a_0, b_1, b_2, \ldots, b_4, c_1, \ldots, c_7$ are values related to the system parameters and observer parameters, which are detailed in Appendix A. It can be seen from the calculation results of Equation (9) that the control input $u$ is not offset in the derivation. This means that the estimated value of the total perturbation $z_4$ is not only related to the parameters of the extended state observer but is also affected by the control input generated by the controller. Therefore, when the value of the control input $u$ changes, even if the total perturbation $F_{Tf}$ does not change, the estimated value of the total perturbation $z_4$ will also fluctuate, which will undoubtedly affect the estimation accuracy of the total perturbation $F_{Tf}$.

For the traditional ESO, the only information used is the position state $x_1$ of the hydraulic servo system. However, the load pressure $x_3$ of the hydraulic servo system can also be measured, and some parameters in the system model can also be known. The full innovation usage or feeding of $x_1$, $x_3$ and the system model is essential to make $z_4$ track $F_{Tf}$ as quickly and accurately as possible.

*2.3. Improvement Extended State Observerwith Model Information and Load Pressure*

To solve the problem of inaccurate perturbation estimation for the hydraulic position servo system by traditional linear ESO, we introduced the hydraulic servo system model into ESO and proposed a new ESO structure, which takes the load pressure as the third input outside the control and system position. This paper gives a finite-time convergence ESO with the load pressure and model information and analyzes its convergence time.

When the total perturbation of the hydraulic systems cannot be described mathematically, the following model-assisted ESO based on model (5) for the hydraulic servo systems can be designed to improve the perturbation estimation performance.

$$\begin{aligned} \dot{z}_{m1} &= z_{m2} - \beta_{m1}(z_{m1} - y) \\ \dot{z}_{m2} &= -l_1 z_{m1} - l_2 z_{m2} + l_3 z_{m3} - \beta_{m2}(z_{m1} - y) \\ \dot{z}_{m3} &= -s_2 z_{m2} - s_3 z_{m3} + z_{m4} + b_0 u - \beta_{m3}(z_{m1} - y) \\ \dot{z}_{m4} &= -\beta_{m4}(z_{m1} - y) \end{aligned} \tag{10}$$

where $\beta_{m1}, \beta_{m2}, \beta_{m3}, \beta_{m4}$ are coefficients to be designed to make ESO Hurwitz stable. The Laplace transformation of formula (10) can be written in the following form:

$$\begin{aligned} z_{m1}(s) &= [1,0,0,0](sI - A_m)^{-1}[Bet_m Y_p(s) + Bu(s)] \\ z_{m2}(s) &= [0,1,0,0](sI - A_m)^{-1}[Bet_m Y_p(s) + Bu(s)] \\ z_{m3}(s) &= [0,0,1,0](sI - A_m)^{-1}[Bet_m Y_p(s) + Bu(s)] \\ z_{m4}(s) &= [0,0,0,1](sI - A_m)^{-1}[Bet_m Y_p(s) + Bu(s)] \end{aligned} \tag{11}$$

where $A_m = \begin{bmatrix} -\beta_{m1} & 1 & 0 & 0 \\ -(\beta_{m2} + l_1) & -l_2 & l_3 & 0 \\ -\beta_{m3} & -s_2 & -s_3 & 1 \\ -\beta_{m4} & 0 & 0 & 0 \end{bmatrix}$, $Bet_m = [\beta_{m1}, \beta_{m2}, \beta_{m3}, \beta_{m4}]^T$, $B = [0, 0, b_0, 0]^T$.

Replacing $Y_p(s)$ in $z_{m4}$ of Formula (11) with $Y_p(s)$ in Formula (8), we can infer that the transfer function of the perturbation observation $z_{m4}$ to the total perturbation $F_{Tf}$ is

$$\frac{z_{m4}(s)}{F_{Tf}(s)} = \frac{b_{m0}}{a_{m4}s^4 + a_{m3}s^3 + a_{m2}s^2 + a_{m1}s + a_{m0}} \tag{12}$$

where $b_{m0}, a_{m0}, \ldots, a_{m4}$ are values related to the system parameters and observer parameters, which are detailed in Appendix A. Compared with Formula (9), Formula (12) has excellent changes, and the transfer function (12) between the total perturbation and its estimated value is only related to the system and observer parameters and is no longer related to the control input. This means that the ESO assisted by the hydraulic system model eliminates the influence of the control input on perturbation observations and has better observation accuracy than traditional ESO.

Notably, in addition to the hydraulic system model, other information can be used in ESO design. The hydraulic system state $x_3$ is usually measurable through the two pressure sensors of two cavities of hydraulic cylinders. This information is quite essential to employ for perturbation observation. It can be seen from Formula (3) that whether it is the total perturbation $F_f$ or the unmodeled parts $\Delta$, their impact on the hydraulic system first acts on the load pressure and then affects the other system states along with the chain structure of the hydraulic system. Therefore, a new-structure linear ESO for a hydraulic system (LHYESO) is proposed by introducing the system state load pressure based on the traditional ESO assisted by the hydraulic system model.

$$\begin{aligned} \dot{z}_{l1} &= z_{l2} - \beta_{l1}(z_{l1} - y) \\ \dot{z}_{l2} &= -l_1 z_{l1} - l_2 z_{l2} + l_3 z_{l3} - \beta_{l2}(z_{l1} - y) \\ \dot{z}_{l3} &= -s_2 z_{l2} - s_3 z_{l3} + z_{l4} + b_0 u - \beta_{l3}(z_{l3} - P_L) \\ \dot{z}_{l4} &= -\beta_{l4}(z_{l3} - P_L) \end{aligned} \tag{13}$$

where $\beta_{l1}, \beta_{l2}, \beta_{l3}, \beta_{l4}$ are coefficients to be designed to make ESO Hurwitz stable. The Laplace transformation of formula (13) can be written in the following form:

$$\begin{aligned} z_{l1}(s) &= [1, 0, 0, 0](sI - A_l)^{-1}[Bet_{l12}Y_p(s) + Bet_{l34}P_L(s) + Bu(s)] \\ z_{l2}(s) &= [0, 1, 0, 0](sI - A_l)^{-1}[Bet_{l12}Y_p(s) + Bet_{l34}P_L(s) + Bu(s)] \\ z_{l3}(s) &= [0, 0, 1, 0](sI - A_l)^{-1}[Bet_{l12}Y_p(s) + Bet_{l34}P_L(s) + Bu(s)] \\ z_{l4}(s) &= [0, 0, 0, 1](sI - A_l)^{-1}[Bet_{l12}Y_p(s) + Bet_{l34}P_L(s) + Bu(s)] \end{aligned} \tag{14}$$

where $A_l = \begin{bmatrix} -\beta_{l1} & 1 & 0 & 0 \\ -(\beta_{l2} + l_1) & -l_2 & l_3 & 0 \\ 0 & -s_2 & -(s_3 + \beta_{l3}) & 1 \\ 0 & 0 & -\beta_{l4} & 0 \end{bmatrix}$, $\begin{matrix} Bet_{l12} = [\beta_{l1}, \beta_{l2}, 0, 0]^T \\ Bet_{l34} = [0, 0, \beta_{l3}, \beta_{l4}]^T \end{matrix}$.

Replacing $Y_p(s)$ and $P_L(s)$ in $z_{l4}(s)$ of Formula (14) with $Y_p(s)$ and $P_L(s)$ in Formula (8), we can infer that the transfer function of the perturbation observation $z_{l4}$ to the total perturbation $F_{Tf}$ is:

$$\frac{z_{l4}(s)}{F_{Tf}(s)} = \frac{b_{l2}s^2 + b_{l1}s + b_{l0}}{s^4 + a_{l3}s^3 + a_{l2}s^2 + a_{l1}s + a_{l0}} \tag{15}$$

where $b_{l0}, \ldots, b_{l2}$ and $a_{l0}, \ldots, a_{l3}$ are values related to the system and observer parameters, detailed in Appendix A.

By comparing the transfer functions between the disturbance estimates and the disturbances of the two structural extended state observers shown in (12) and (15), it was found that their denominator parts have the same structure, and the highest order front

coefficients are one, and both have constant terms. On satisfying stability, they all have four poles in the left half of the complex plane. Therefore, the denominator part of the two formulas can be seen as a series of four integral links, which is equivalent to the two ESOs which have four low-pass filters in the perturbation estimation channel.

However, the transfer function of (15) is two zeros more than that of (12). This shows that the LHYESO weakens the effect of the low-pass filtering of model-assisted ESO, which makes the turning point of the logarithmic frequency curve move to the right, and the advantage is to increase the bandwidth of the perturbation observation channel. To visually observe the frequency domain characteristics of the transfer functions (12) and (15), we draw the Bode diagram of the two transfer functions shown in Figure 2, according to the hydraulic system and the ESO parameters shown in Table 1.

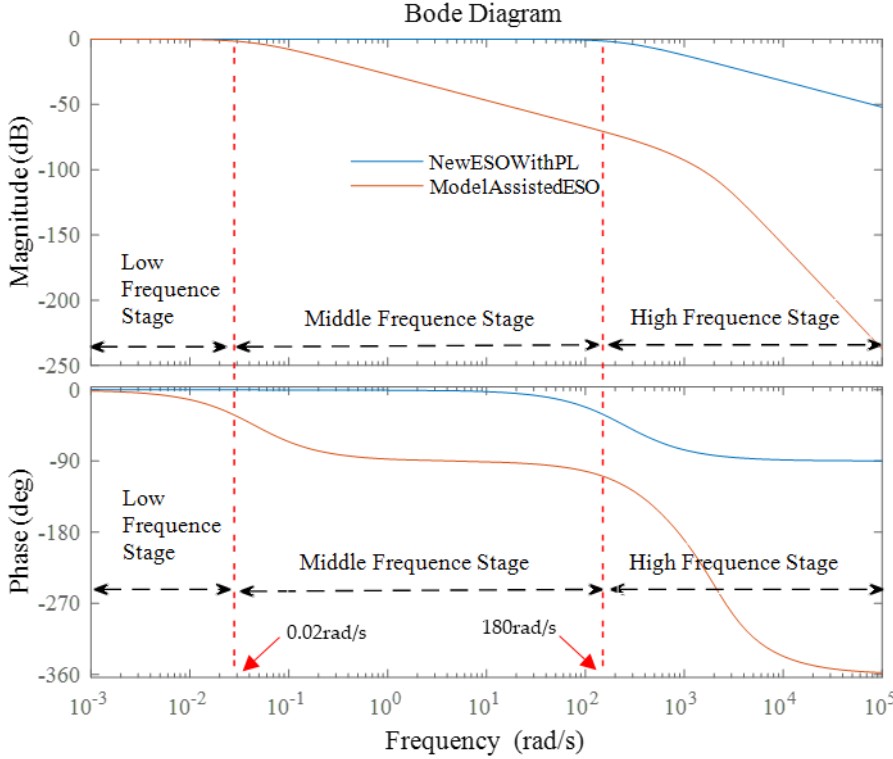

**Figure 2.** The bode plots of transfer functions between perturbation estimation and perturbation for two structure ESOs.

**Table 1.** Parameters of the hydraulic system and ESOs.

| Parameters | Value | Parameters | Value |
|---|---|---|---|
| $A(\mathrm{m}^2)$ | $3.4 \times 10^{-3}$ | $\beta_{m1}$ | $4 \times 10^3$ |
| $m_{load}(\mathrm{kg})$ | 25 | $\beta_{m2}$ | $6 \times 10^6$ |
| $B_p(\mathrm{N(m/s)})$ | 200 | $\beta_{m3}$ | $4 \times 10^9$ |
| $V_t(\mathrm{m}^3)$ | $1.72 \times 10^{-3}$ | $\beta_{m4}$ | $1 \times 10^{12}$ |
| $\beta_e(\mathrm{P_a})$ | $6.9 \times 10^8$ | $\beta_{l1}$ | $4 \times 10^3$ |
| $K_c(\mathrm{m}^3/(\mathrm{s.p_a}))$ | $3.58 \times 10^{-12}$ | $\beta_{l2}$ | $6 \times 10^6$ |
| $C_{tp}(\mathrm{m}^3/(\mathrm{s.p_a}))$ | $1 \times 10^{-19}$ | $\beta_{l3}$ | $4 \times 10^9$ |
| $K_s(\mathrm{N/m})$ | 0 | $\beta_{l4}$ | $1 \times 10^{12}$ |
| $K_q(\mathrm{m}^2/\mathrm{s})$ | 1.67 | | |
| $K_{sv}(\mathrm{m/mA})$ | $1.25 \times 10^{-5}$ | | |

For the convenience of description, we artificially divided the bode diagram into low-frequency, middle-frequency, and high-frequency stages. From the low-frequency stage, we could see that the logarithmic amplitude–frequency characteristic curves of the

two ESOs coincided with the zero decibels. The phase–frequency characteristic curve of the new structure ESO almost coincides with the 0–degree curve. It is higher than the model-assisted ESO, but the latter is between 0 and −40 degrees. This shows that both ESOs can reasonably estimate the perturbation in the low-frequency band, but the estimation of the perturbation by the model-assisted ESO is slightly lagging. In the middle frequency band, two kinds of ESOs have a significant difference in the performance of perturbation estimation. The amplitude–frequency characteristic curve of the new structure ESO can still maintain the coincidence with 0 decibels, and its phase–frequency characteristic curve decreases in the middle frequency band but can still be greater than −40 degrees. However, the amplitude–frequency characteristic curve of model-assisted ESO begins to decrease with the slope of −20 dB per ten octaves, and its phase–frequency characteristic curve is between −40 degrees and −90 degrees. This shows that the new structure ESO is superior to the model-assisted ESO in estimating the intermediate frequency perturbation of the hydraulic system. The LHYESO and the model-assisted ESO trend are the same in the high-frequency band. However, the amplitude and phase characteristic curves of the former are higher than those of the latter, which shows that the new ESO is better than the model-assisted ESO for the high-frequency perturbation of the hydraulic system. The maximum effective estimated frequency of the ESO with the new structure for high-frequency perturbation is about 180 rad/s.

The results in Figure 2 are consistent with the analysis results. The LHYESO weakens the effect of a series connection of multiple low-pass filters by increasing the zero point, which expands the perturbation estimation frequency range. Although we can adjust the perturbation observation effect by adjusting the parameters of the model-assisted ESO in the simulation, in the practical field, the excessive gain will cause the system to be too sensitive to noise and endanger the system's stability. Under the same observer gain, the estimation range of the disturbance frequency of LHYESO is broader than the model-assisted ESOs, and the model-aided observer needs higher gains than LHYESO to realize the same perturbation estimation effects.

### 2.4. Finite-Time Extended State Observer with Model Information and Load Pressure

In order to improve the estimation speed for hydraulic systems states and disturbance, we further design a finite time convergent ESO for hydraulic systems (FTHYESO) based on the structure of the (13). To express clearly, the following lemans used in this paper are introduced.

**Lemma 1.** [42,43]. *Consider the system of state equations*

$$\dot{x}_a(t) = f(x_a(t)) \tag{16}$$

*where* $f : \mathcal{D} \to \mathbb{R}^n$ *is continuous on an open neighbourhood* $\mathcal{D} \subseteq \mathbb{R}^n$ *of the origin and* $f(0) = 0$. *Suppose there exists a continuous function* $V : \mathcal{D} \to \mathbb{R}$ *such that the following condition hold: V is positive definite. Then, there exists* $c > 0$, $\varrho \in (0, 1)$ *and an open neighbourhood* $\varpi \subseteq \mathcal{D}$ *of the origin such that*

$$\dot{V}_a(x_a(t)) + c(V_a(x_a(t)))^Q \leq 0, \ x_a(t) \in \varpi \smallsetminus \{0\} \tag{17}$$

Then, the origin is a finite-time-stable equilibrium of (16). The setting-time function $T$ is shown as follows:

$$T_1 \leq \frac{1}{c(1-\varrho)} (V_a(x_a(t_0)))^{1-\varrho} \tag{18}$$

where $V_a(x_a(t_0))$ is the initial value of $V_a(x_a(t))$ and $T$ is continuous.

Based on (13), an FTHYESO is presented as follows:

$$\begin{cases} \dot{z}_{n1} = z_{n2} - \beta_{n1}\text{sig}^{(\alpha+1)/2}(z_{n1} - x_1) \\ \dot{z}_{n2} = \frac{1}{m_{load}}(Az_{n3} - B_p z_{n2} - K_s z_{n1}) - \beta_{n2}\text{sig}^{(\alpha+1)/2}(z_{n1} - x_1) \\ \dot{z}_{n3} = \frac{4\beta_e}{V_t}(K_q K_{sv} u - Az_{n2} - C_{tp} z_{n3}) + z_{n4} - \beta_{n3}\text{sig}^{(\alpha+1)/2}(z_{n3} - x_3) \\ \dot{z}_{n4} = -\beta_{n4}\text{sig}^{(\alpha+1)/2}(z_{n3} - x_3) \end{cases} \quad (19)$$

where $\text{sig}^{(\alpha+1)/2}(x) = \text{sgn}(x)|x|^{(\alpha+1)/2}$, $\text{sgn}()$ is the standard symbolic function, $\alpha$ is the given parameter with $0 < \alpha < 1$, and $z_{n1}, \ldots, z_{n4}$ are the observation values for the system state $x_1, \ldots, x_4$ of the system (5).

Considering FTHYESO (19) and system (5), we obtain the error function as follows:

$$\begin{cases} \dot{e}_1 = e_2 - \beta_{n1}\text{sig}^{(\alpha+1)/2}(e_1) \\ \dot{e}_2 = -l_1 e_1 - l_2 e_2 + l_3 e_3 - \beta_{n2}\text{sig}^{(\alpha+1)/2}(e_1) \\ \dot{e}_3 = -s_2 e_2 - s_3 e_3 + e_4 - \beta_{n3}\text{sig}^{(\alpha+1)/2}(e_3) \\ \dot{e}_4 = -\beta_{n4}\text{sig}^{(\alpha+1)/2}(e_3) - \dot{F}_{Tf} \end{cases} \quad (20)$$

where $e_1 = z_{n1} - x_1, \ldots, e_4 = z_{n4} - x_4$, $l_1, l_2, l_3, s_2, s_3$ can be found in (5). The Lyapunov theory is employed to analyze the observation errors as follows.

An appropriate Lyapunov function is constructed as follows:

$$V = \zeta^T P\zeta = \frac{2\beta_{n1}}{\alpha+1}|e_1|^{\alpha+1} + e_2^2 + \frac{\beta_{n3}}{\alpha+1}|e_3|^{\alpha+1} + \beta_{n4}e_4^2 \\ + [-l_1 e_1 - l_2 e_2 + l_3 e_3 - \beta_{n2}sig^{(\alpha+1)/2}(e_1)]^2 + [-s_2 e_2 - s_3 e_3 + e_4 - \beta_{n3}sig^{(\alpha+1)/2}(e_3)]^2 \geq 0 \quad (21)$$

where $\zeta^T = [|e_1|^{(\alpha+1)/2}\text{sgn}(e_1), e_1, e_2, |e_3|^{(\alpha+1)/2}\text{sgn}(e_3), e_3, e_4]^T$; $P$ is in Appendix A. When $e_1 \neq 0$, the Lyapunov function $V$ is continuous and differentiable. The $\dot{\zeta}$ can be obtained as:

$$\dot{\zeta} = \begin{bmatrix} \frac{\alpha+1}{2}|e_1|^{\frac{\alpha-1}{2}}[e_2 - \beta_{n1}\text{sig}^{(\alpha+1)/2}(e_1)] \\ e_2 - \beta_{n1}\text{sig}^{(\alpha+1)/2}(e_1) \\ -l_1 e_1 - l_2 e_2 + l_3 e_3 - \beta_{n3}\text{sig}^{(\alpha+1)/2}(e_1) \\ \frac{\alpha+1}{2}|e_3|^{\frac{\alpha-1}{2}}[-s_2 e_2 - s_3 e_3 + e_4 - \beta_{n3}\text{sig}^{(\alpha+1)/2}(e_3)] \\ -s_2 e_2 - s_3 e_3 + e_4 - \beta_{n3}\text{sig}^{(\alpha+1)/2}(e_3) \\ -\beta_{n4}\text{sig}^{(\alpha+1)/2}(e_3) - \dot{F}_{Tf} \end{bmatrix} = A_\zeta \zeta - B_f \dot{F}_{Tf}$$

$$= \begin{bmatrix} -k\mu_1\beta_{n1} & 0 & k\mu_1 & 0 & 0 & 0 \\ -\beta_{n1} & 0 & 1 & 0 & 0 & 0 \\ -\beta_{n2} & -l_1 & -l_2 & 0 & l_3 & 0 \\ 0 & 0 & -k\mu_3 s_2 & -k\mu_3\beta_{n3} & -k\mu_3 s_3 & k\mu_3 \\ 0 & 0 & -s_2 & -\beta_{n3} & -s_3 & 1 \\ 0 & 0 & 0 & -\beta_{n4} & 0 & 0 \end{bmatrix} \begin{bmatrix} \text{sig}^{(\alpha+1)/2}(e_1) \\ e_1 \\ e_2 \\ \text{sig}^{(\alpha+1)/2}(e_3) \\ e_3 \\ e_4 \end{bmatrix} - \begin{bmatrix} 0 \\ 0 \\ 0 \\ 0 \\ 0 \\ 1 \end{bmatrix} \dot{F}_{Tf} \quad (22)$$

where $1/2 < k = (\alpha+1)/2 < 1$, $\mu_1 = |e_1|^{(\alpha-1)/2} > 0$, $\mu_3 = |e_3|^{(\alpha-1)/2} > 0$; the rational design coefficients $\beta_{n1}, \ldots, \beta_{n4}$ ensure that matrix $A_\zeta$ is Hurwitz stable.

The time differential equation of $V$ is expressed as follows:

$$\dot{V} = \dot{\zeta}^T P\zeta + \zeta^T P\dot{\zeta} = \zeta^T(A^T P + PA)\zeta - 2B_f^T P\zeta\dot{F}_{Tf} = -\zeta^T Q\zeta - 2B_f^T P\zeta\dot{F}_{Tf} \quad (23)$$

where $Q = -(A^T P + PA) > 0$ is the positive definite matrix.

According to (21), the following inequality is obtained as:

$$\lambda_{\min}\{P\}\|\zeta\|_2^2 \leq V \leq \lambda_{\max}\{P\}\|\zeta\|_2^2 \quad (24)$$

where $\|\zeta\|_2^2 = |e_1|^{\alpha+1} + e_2^2 + |e_3|^{\alpha+1} + e_3^2 + e_4^2$. Hence, the following inequality can be obtained as:

$$\|\zeta\|_2^2 \geq |e_1|^{\alpha+1} \quad (25)$$

According to (23), the inequality regarding the derivative of $V$ can be obtained as follows:

$$\dot{V} \leq -\lambda_{\min}\{Q\}\|\zeta\|_2^2 + 2L_dF_d\|\zeta\|_2 = -(\lambda_{\min}\{Q\}\|\zeta\|_2 - 2L_dF_d)\|\zeta\|_2 \qquad (26)$$

where $L_d = \|-B_f^T P\|_2 = (s_2^2 + \beta_{n3}^2 + s_3^2 + (\beta_{n4}^2 + 1)^2)^{1/2}$.

For the positive definite matrix $Q$ composed of real numbers, whose minimum singular value $\delta_{\min}\{Q\}$ is equal to the minimum eigenvalue $\lambda_{\min}\{Q\}$. According to (23) the following inequality can be obtained as:

$$\sigma_{\min}\{Q\} = 2\sigma_{\min}\{-AP\} \geq 2\sigma_{\min}\{-A\}\sigma_{\min}\{P\} \qquad (27)$$

The matrix $A$ is rewritten as:

$$-A = \begin{bmatrix} k\mu_1 & 0 & 0 & 0 & 0 & 0 \\ 0 & 1 & 0 & 0 & 0 & 0 \\ 0 & 0 & 1 & 0 & 0 & 0 \\ 0 & 0 & 0 & k\mu_3 & 0 & 0 \\ 0 & 0 & 0 & 0 & 1 & 0 \\ 0 & 0 & 0 & 0 & 0 & 1 \end{bmatrix} \begin{bmatrix} \beta_{n1} & 0 & -1 & 0 & 0 & 0 \\ \beta_{n1} & 0 & -1 & 0 & 0 & 0 \\ \beta_{n2} & l_1 & l_2 & 0 & -l_3 & 0 \\ 0 & 0 & s_2 & \beta_{n3} & s_3 & -1 \\ 0 & 0 & s_2 & \beta_{n3} & s_3 & -1 \\ 0 & 0 & 0 & \beta_{n4} & 0 & 0 \end{bmatrix} = A_1 A_2 \qquad (28)$$

According to (27) and (28), the following inequalities are obtained as:

$$\sigma_{\min}\{Q\} \geq 2\sigma_{\min}\{-A\}\sigma_{\min}\{P\} = 2\sigma_{\min}\{A_1 A_2\}\sigma_{\min}\{P\} \geq 2\sigma_{\min}\{A_1\}\sigma_{\min}\{A_2\}\sigma_{\min}\{P\} \qquad (29)$$

The initial observation error of the observer is usually large and usually small when it converges to the actual state value. According to the trend of the error variation, $\sigma_{\min}\{A_1\}$ is rewritten as:

$$\sigma_{\min}\{A_1\} = \begin{cases} 1, & e_1 \leq e_3 < (\frac{2}{\alpha+1})^{2/(\alpha-1)} \\ k\mu_1, & e_1 \geq e_3 \geq (\frac{2}{\alpha+1})^{2/(\alpha-1)} \end{cases} \qquad (30)$$

According to (25), (29), and (30), when $e_1 \geq (2/(\alpha+1))^{2/(\alpha-1)}$, the relationship $\|\zeta\|_2 \geq |e_1|^{(\alpha+1)/2} \geq (2/(\alpha+1))^{(\alpha+1)/(\alpha-1)}$ is established, and the following inequalities relationship is obtained as:

$$\begin{aligned} q_{d1} &= \lambda_{\min}\{Q\}\|\zeta\|_2 - 2L_dF_d \\ &\geq 2k\mu_1\sigma_{\min}\{A_2\}\sigma_{\min}\{P\}|e_1|^{(\alpha+1)/2} - 2L_dF_d \\ &= (\alpha+1)\sigma_{\min}\{A_2\}\sigma_{\min}\{P\}|e_1|^{\alpha} - 2L_dF_d \\ &\geq (\alpha+1)(\frac{2}{\alpha+1})^{2\alpha/(\alpha-1)}\sigma_{\min}\{A_2\}\sigma_{\min}\{P\} - 2L_dF_d \\ &\geq 0.7358 \times \sigma_{\min}\{A_2\}\sigma_{\min}\{P\} - 2L_dF_d \\ &\triangleq q_{d1,\min} \end{aligned} \qquad (31)$$

To satisfy the inequality relationship, $q_{d_1,\min} > 0$, $\beta_{n1}, \ldots, \beta_{n4}$ are adjusted by carrying out corresponding experiments. Considering (24), (26), and (31), it follows that:

$$\dot{V} \leq -q_{d1}\|\zeta\|_2 \leq -\sqrt{\frac{q_{d1,\min}^2 V}{\lambda_{\max}\{P\}}} = -T_1\sqrt{V} < 0 \qquad (32)$$

According to Lemma 1, when the observation error of FTHYESO converges to $\|\zeta\|_2 < (2/(\alpha+1))^{(\alpha+1)/(\alpha-1)}$ in finite-time, the adjusted time $t_1$ is written as follows:

$$t_1 \leq \frac{2}{T_1}\sqrt{V} \leq \frac{2}{T_1}\sqrt{V(0)} \qquad (33)$$

where $V(0)$ is the initial value of $V$ under the initial observation error.

When $\|\zeta\|_2 < (2/(\alpha+1))^{(\alpha+1)/(\alpha-1)}$, then $e_1 \leq (2/(\alpha+1))^{2/(\alpha-1)}$. According to (25), (29), and (30), there exists:

$$
\begin{aligned}
q_{d2} &= \lambda_{\min}\{Q\}\|\zeta\|_2 - 2L_d F_d \\
&\geq 2\sigma_{\min}\{A_2\}\sigma_{\min}\{P\}\|\zeta\|_2 - 2L_d F_d
\end{aligned}
\tag{34}
$$

When the following relationship is satisfied:

$$
\left(\frac{2}{(\alpha+1)}\right)^{(\alpha+1)/(\alpha-1)} > \|\zeta\|_2 > \frac{L_d F_d}{\sigma_{\min}\{A_2\}\sigma_{\min}\{P\}}
\tag{35}
$$

Then, there exists $q_{d2} > 0$. According to (24), (26), and (35), the following inequality is obtained as:

$$
\dot{V} \leq -q_{d2}\|\zeta\|_2 \leq -\sqrt{\frac{q_{d2}^2 V}{\lambda_{\max}\{P\}}} = -T_2\sqrt{V} < 0
\tag{36}
$$

According to Lemma 1, the adjusted time $t_1$ is written as follows:

$$
t_2 \leq \frac{2}{T_2}\sqrt{V} \leq \frac{2}{T_2}\sqrt{V(t_1)}
\tag{37}
$$

Finally, the observation error converges to the following equation in finite time $t = t_1 + t_2$ as:

$$
\|\zeta\|_2 < \frac{L_d F_d}{\sigma_{\min}\{A_2\}\sigma_{\min}\{P\}}
\tag{38}
$$

Obviously, selecting the appropriate observer parameters $\beta_{n1}, \ldots, \beta_{n4}, \sigma_{\min}\{A_2\}\sigma_{\min}\{P\}$ need to be large enough to ensure that the error eventually converges to a small value. The finite-time convergent ESO designed for the hydraulic system is proven.

*2.5. Finite-Time Backstepping Controller Design*

Before the controller design, a critical lemma is given as follows.

**Lemma 2.** [44,45]. *Suppose $a_1, a_2, \ldots a_n$ and $0 < num < 2$ are all real numbers. Then, the following inequality holds:*

$$
|a_1|^{num} + |a_2|^{num} + \ldots + |a_n|^{num} \geq \left(a_1^2 + a_2^2 + \ldots + a_n^2\right)^{num/2}
\tag{39}
$$

Based on the excellent state estimation by FTHYESO (19), a backstepping controller is designed in this section to ensure the good performance for the hydraulic servo system (5). The tracking error is defined as follows:

$$
\begin{cases}
\tau_1 = x_1 - y_d \\
\tau_2 = x_2 - x_{2d} \\
\tau_3 = x_3 - x_{3d}
\end{cases}
\tag{40}
$$

where $y_d$ is the desired position input signal, and $x_{2d}$ and $x_{3d}$ are both the virtual control signals that need further calculation.

The Lyapunov function $V_1$ can be defined as follows:

$$
V_1 = \frac{1}{2}\tau_1^2
\tag{41}
$$

The time differential equation of $V_1$ is expressed as follows:

$$
\begin{aligned}
\dot{V}_1 &= \tau_1\dot{\tau}_1 \\
&= \tau_1(\dot{x}_1 - \dot{y}_d) \\
&= \tau_1(x_2 - \dot{y}_d) \\
&= \tau_1(\tau_2 + x_{2d} - \dot{y}_d)
\end{aligned}
\tag{42}
$$

The virtual control variable $x_{2d}$ can be constructed as follows:

$$x_{2d} = \dot{y}_d - k_1 \tau_1^{2\beta-1} \tag{43}$$

where $k_1$ is the positive parameter to be designed, $\beta$ is the given parameter with $0 < \beta < 1$. Then, considering (42) and (43), the $\dot{V}_1$ is obtained as:

$$\dot{V}_1 = \tau_1 \tau_2 - k_1 \tau_1^{2\beta} \tag{44}$$

To go one step ahead, a new Lyapunov function $V_2$ is defined as:

$$V_2 = V_1 + \frac{1}{2}\tau_2^2 \tag{45}$$

By taking the derivation of Equation (45)

$$\begin{aligned}
\dot{V}_2 &= \dot{V}_1 + \tau_2 \dot{\tau}_2 \\
&= \dot{V}_1 + \tau_2(\dot{x}_2 - \dot{x}_{2d}) \\
&= \dot{V}_1 + \tau_2(-l_1 x_1 - l_2 x_2 + l_3 \tau_3 + l_3 x_{3d} - \dot{x}_{2d})
\end{aligned} \tag{46}$$

Then, the virtual control variable $x_{3d}$ can be designed as:

$$x_{3d} = \frac{1}{l_3}(l_1 x_1 + l_2 x_2 + \dot{x}_{2d} - \tau_1 - k_2 \tau_2^{2\beta-1}) \tag{47}$$

where the new parameter $k_2$ is the positive parameter to be designed. The (45) can be rewritten as:

$$\begin{aligned}
\dot{V}_2 &= \dot{V}_1 + \tau_2(l_3 \tau_3 - \tau_1 - k_2 \tau_2^{2\beta-1}) \\
&= -k_1 \tau_1^{2\beta} - k_2 \tau_2^{2\beta} + l_3 \tau_2 \tau_3
\end{aligned} \tag{48}$$

The Lyapunov function $V_3$ can be defined as follows:

$$V_3 = V_2 + \frac{1}{2}\tau_3^2 \tag{49}$$

The time differential equation of $V_3$ is expressed as follows:

$$\begin{aligned}
\dot{V}_3 &= \dot{V}_2 + \tau_3 \dot{\tau}_3 \\
&= \dot{V}_2 + \tau_3(\dot{x}_3 - \dot{x}_{3d}) \\
&= \dot{V}_2 + \tau_3(-s_2 x_2 - s_3 x_3 + x_4 + b_0 u - \dot{x}_{3d})
\end{aligned} \tag{50}$$

To make sure $\dot{V}_3 < 0$, the control variable $u$ can be designed as:

$$\begin{aligned}
u &= \frac{1}{b_0}(s_2 x_2 + s_3 x_3 - x_4 + \dot{x}_{3d} - l_3 \tau_2 - k_3 \tau_3^{2\beta-1}) \\
&= \frac{1}{b_0}(s_2 x_2 + s_3 x_3 - x_4 + \frac{1}{l_3}(l_1 x_2 + l_2(-l_1 x_1 - l_2 x_2 + l_3 x_3) \\
&\quad + \ddot{y}_d - k_1(\tau_1^{2\beta-1})'' - \dot{\tau}_1 - k_2(\tau_2^{2\beta-1})') - l_3 \tau_2 - k_3 \tau_3^{2\beta-1})
\end{aligned} \tag{51}$$

According to the (50) and (51), the $\dot{V}_3$ is given as:

$$\dot{V}_3 = -k_1 \tau_1^{2\beta} - k_2 \tau_2^{2\beta} - k_3 \tau_3^{2\beta} \tag{52}$$

where $k_1 > 0, k_2 > 0, k_3 > 0$, and $0 < \beta < 1$. According to Lemma 2, one has

$$\begin{aligned}
\dot{V}_3 &\leq -k_{\min}(\tau_1^{2\beta} + \tau_2^{2\beta} + \tau_3^{2\beta}) \\
&\leq -\frac{1}{2}^{-\beta} k_{\min}(\frac{1}{2}\tau_1^2 + \frac{1}{2}\tau_2^2 + \frac{1}{2}\tau_3^2)^{\frac{\beta}{2}} \\
&= -k V_3^{\frac{\beta}{2}}
\end{aligned} \tag{53}$$

where $k = (1/2)^{-\beta} k_{\min}, k_{\min} = \min\{k_1, k_2, k_3\}$.

By using Lemma 1, it can be concluded the tracking error of the hydraulic servo system (5) converges to the equilibrium point in finite time despite external disturbance and system uncertainty.

## 3. Results and Discussions

### 3.1. Experiment Platform

The experiment platform of the hydraulic servo system is shown in Figure 3. The load simulation system on the left side of Figure 3 is to be applied to simulated loads in the electro-hydraulic position servo system during a motion to verify the effectiveness of the designed controller and observer. The experiment platform consists of seven main parts: a hydraulic rod cylinder (SFQ80/45-500, no-load friction: ≤0.05 MPa, Manufacturer information: WeiHang, Shandong, China), a servo valve (FF130/60, frequency characteristics: ≥100 Hz (−3 dB), hysteresis: ≤4%, Manufacturer information: NJSERVO, Nanjing, China), two pressure sensors (KYB20, full scale: 21 MPa, nonlinear error: 0.3% F.S, Manufacturer information: KangYu, Guangdong Province, China), a displacement sensor (DM-L500, full scale: 500 mm, nonlinear error: 0.05% F.S, Manufacturer information: KangYu, Guangdong Province, China), an analog acquisition/output card (PCI-1711, output ranges: 0~10 V, resolution: 12-bit, Manufacturer information: Advantech, Suzhou Province, China), an industrial control computer (610L, Manufacturer information: Advantech, Suzhou Province, China), and an external load force loading system (Maximum Loading Force: 50,000 N, Force Sensor: CAAA, BK1B-5t, Full scale: 50,000 N, Nonlinear error: 0.05% F.S). The controller is built with the 5.9 community version of the Qt Company and loads the Qtmath library and Advantech PC1-1711 analog acquisition and output library functions.

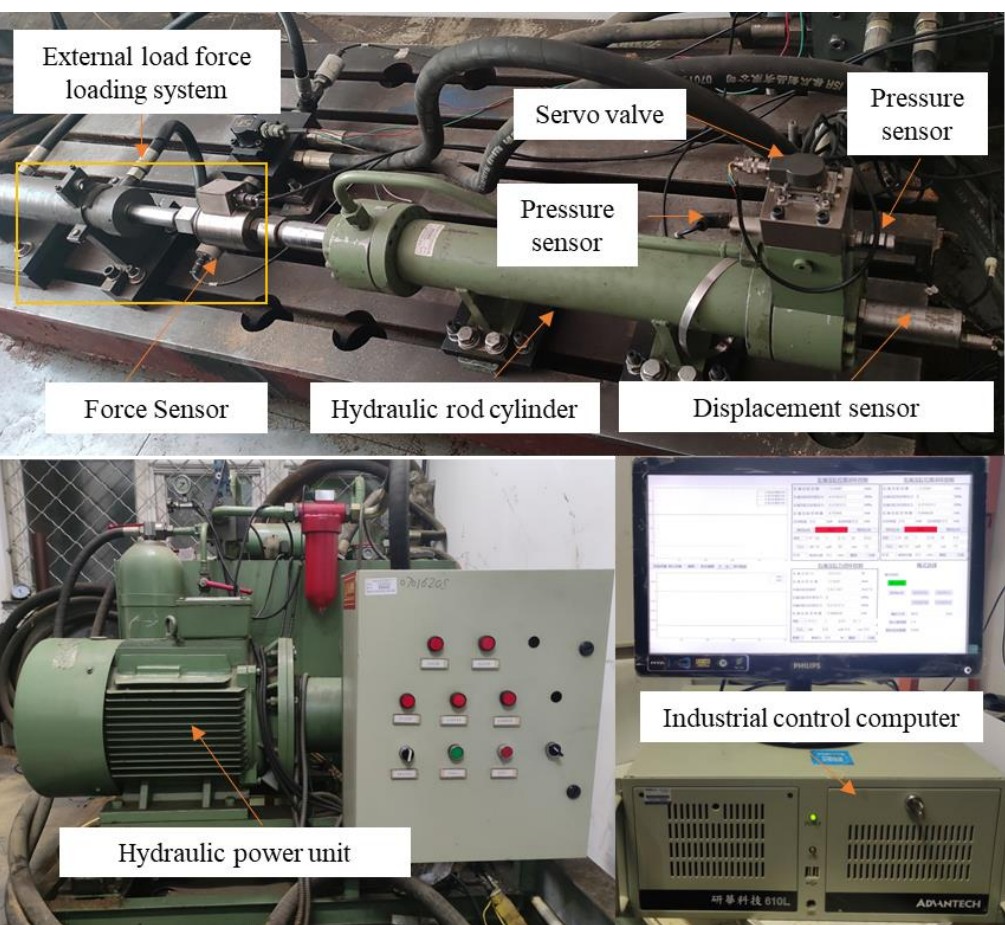

**Figure 3.** The platform of the hydraulic servo system.

### 3.2. Results and Analyses

To verify the transient character, in the first experiment, we compared the control performance of the backstepping controller basing FTHYESO and the linear active disturbance rejection control (LADRC) under the tracking of a constant value trajectory of 200 mm. In the second experiment, the sinusoidal trajectory was used to verify the tracking performance, whose frequency was 0.2 Hz, the offset was 150 mm, and the amplitude was 150 mm. The initial conditions of the two experiments are the same, the initial position of the hydraulic cylinder was 0 mm, the initial velocity was 0 mm/s, the initial load pressure was 0 MPa, and the initial control input was 0 mA. To truly verify the performance that the controller proposed in this paper, we applied a nonlinear time-varying load force through the load simulation system in two sets of comparative experiments, whose $F_f = 10,000 \sin(0.4\pi t)$N.

The proposed method's adjustable observer and controller parameters are listed in Table 2, and the experiment results are shown in Figures 4 and 5. Note that $y_d$ is the expected tracking trajectory, I and L are the tracking results by the proposed method and traditional LADRC method, respectively. U − I and U − L are the control inputs of the proposed and traditional LADRC methods, respectively. $Z_4$−I and $Z_4$−L are estimations for the total perturbation by the proposed FTHYESO and the traditional linear ESO, respectively. error − I and error − L are tracking errors in the proposed and the traditional LADRC method, respectively. force − I and force − L are the load forces experienced by the proposed controller and traditional LADRC experiments, respectively.

**Table 2.** Parameters of the proposed method.

| Parameters | Value | Parameters | Value |
|---|---|---|---|
| $\beta_{n1}$ | 400 | $k_1$ | $1.25 \times 10^8$ |
| $\beta_{n2}$ | 60,000 | $k_2$ | $7.5 \times 10^5$ |
| $\beta_{n3}$ | 400 | $k_3$ | 1500 |
| $\beta_{n4}$ | 60,000 | $b_0$ | 3000 |
| $\alpha$ | 0.8 | $\beta$ | 0.9 |

In the first comparative experiment, the transient character of the two controllers showed a difference under almost the same external load force (Figure 4e). Figure 4a illustrates how the position moved to the given tracking position at about 0.7s for both methods, the traditional control method needed about 3.7 s of tuning time to be stable, and the proposed method hardly needed any tuning time. As shown in Figure 4b, both the two control inputs tended to be stable, which made sure that the system (4) could stay at the given constant value trajectory, but it is obvious that one of the proposed methods tended to be stable faster than LADRC. From Figure 4c, it can be seen that both methods could estimate the total perturbation stably, but the adjustment time of the traditional method was about 4 s, and the proposed method almost did not need any adjustment time. In Figure 4d, it shows that the final steady-state error of the two methods was < 0.05 mm, but the overshoot of the proposed method was far less than that of the traditional LADRC.

In the second comparative experiment, the tracking performance of the two controllers showed a difference under almost the same external load force (Figure 5e). Note that the controller and observer parameters of the two methods in the second experiment are the same as those in the first experiment. From Figure 5a, the tracking trajectory of the system (5), based on the FTHYESO and backstepping controller, was closer to the desired trajectory than the LADRC, especially in the peaks and valleys of the set sinusoidal trajectory. In Figure 5b, the control input of the two methods appears close, but the control input of the proposed method was smoother than that of the traditional method. Figure 5c shows the trends of the two methods for the total perturbation estimation of the system (5), and the proposed FTHYESO clearly had a better convergence speed than the TESO. It can be seen from Figure 5d that the tracking error of the proposed method was much smaller than that of the LADRC.

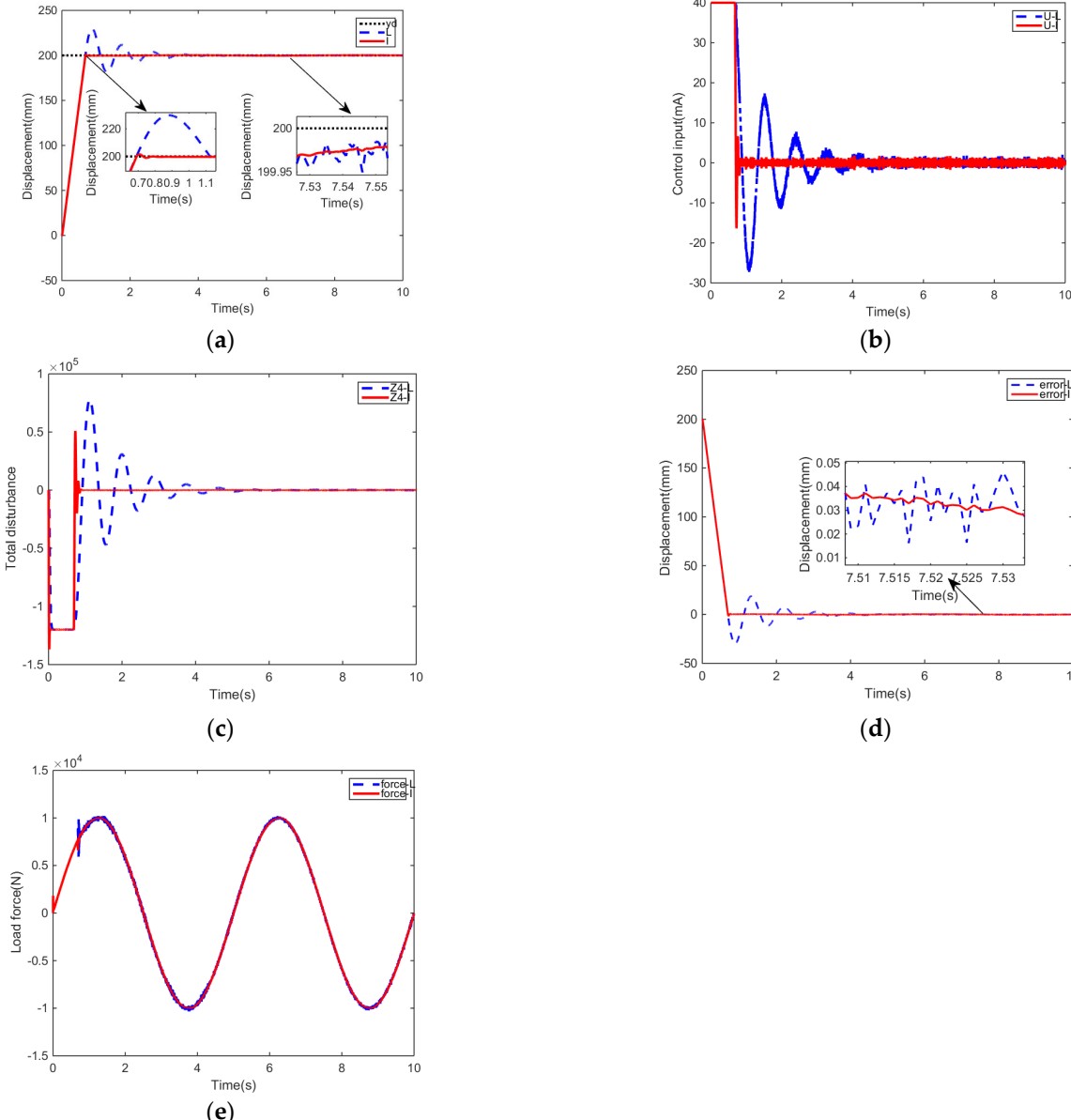

**Figure 4.** Experimental results of the proposed method and the traditional LADRC method for tracking a constant value trajectory of 200 mm; (**a**) Comparison of tracking performance; (**b**) Comparison of control inputs; (**c**) Comparison of perturbation estimation; (**d**) Comparison of tracking error (**e**) External load force.

According to Figures 4 and 5, the comparisons of the experimental results between the two methods are calculated in Table 3. It is evident from Table 3 that the hydraulic servo system (5), based on FTHYESO and the backstepping controller, had superior tracking rapidity and lower tracking error compared with the LADRC.

**Table 3.** Comparison of experimental results.

| Controller | First Experiment | | Second Experiment | |
|---|---|---|---|---|
| | Adjustable Time | Steady-State Error | Delay Time | Average Error |
| LADRC | 4 s | 0.05 mm | 0.1 s | 4 mm |
| This paper | 0.7 s | 0.03 mm | 0.02 s | 0.1 mm |

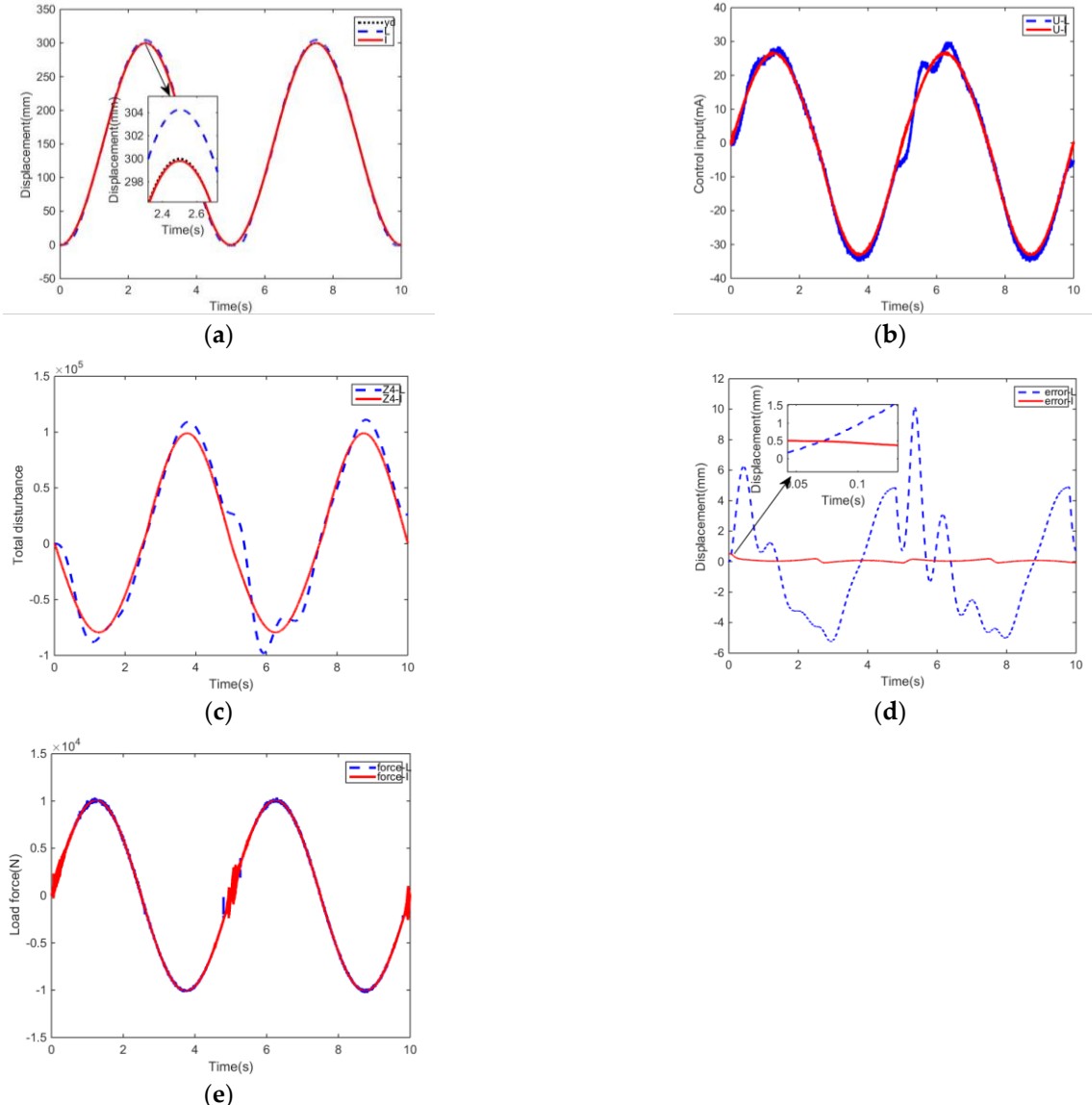

**Figure 5.** Experimental results of the proposed method and the traditional LADRC method for tracking sinusoidal trajectory; (**a**) Comparison of tracking performance; (**b**) Comparison of control inputs; (**c**) Comparison of perturbation estimation; (**d**) Comparison of tracking errors; (**e**) External load force.

## 4. Conclusions

An LHYESO designed with a new structure has been designed in this paper to improve the perturbation estimation performance for the hydraulic system. Compared with TESO, LHYESO employs the model information and load pressure to complete the design of the structure. The corresponding frequency domain analysis results show that it eliminates the control input in the transfer function and reduces the dependence of the high-frequency domain range of the perturbation estimation on the significant observer gain. Then, an FTHYESO with finite-time convergence has been proposed to quickly and accurately observe the syste'sm total perturbation, and the stability analysis of the finite-time convergence in response to a sufficiently small observation error was carried out. Moreover, a finite-time backstepping controller has been designed by the Lyapunov method to guarantee the rapidity and precise response of the hydraulic servo system based on FTHYESO. Finally, static and dynamic tracking experiments have been carried out, respectively. By comparing the experimental results with those of LADRC, the proposed

method in this paper has significantly improved the hydraulic servo system's tracking accuracy and response speed.

**Author Contributions:** Conceptualization, F.M. and H.Y.; methodology, F.M.; software, F.M.; validation, F.M., H.Y., J.L. and X.L.; formal analysis, F.M.; investigation, F.M.; writing—original draft preparation, F.M.; writing—review and editing, F.M. All authors have read and agreed to the published version of the manuscript.

**Funding:** This research was funded by Fundamental Research Funds for the Central Universities, grant number 2021YJS141.

**Data Availability Statement:** Data will be made available from the corresponding author on reasonable request.

**Conflicts of Interest:** The authors declare no conflict of interest.

**Appendix A**

$a_0 = \beta_4 l_3, b_1 = b\beta_4(l_3 - 1), b_2 = b\beta_4(l_2 + s_3), b_3 = b\beta_4(l_1 + l_2 s_3 + l_3 s_2), b_4 = b\beta_4 l_1 s_3$

$c_1 = (\beta_1 + l_2 + s_3), c_2 = (\beta_2 + l_2 + \beta_1 l_2 + \beta_1 s_3 + l_2 s_3 + l_3 s_2)$

$c_3 = (\beta_3 + \beta_1 l_1 + \beta_2 l_2 + \beta_2 s_3 + l_1 s_3 + \beta_1 l_2 s_3 + \beta_1 l_3 s_2)$

$c_4 = (\beta_4 + \beta_2 l_1 + \beta_3 l_2 + \beta_3 s_3 + \beta_1 l_1 s_3 + \beta_2 l_2 s_3 + \beta_2 l_3 s_2)$

$c_5 = (\beta_3 l_1 + \beta_4 l_2 + \beta_4 s_3 + \beta_2 l_1 s_3 + \beta_3 l_2 s_3 + \beta_3 l_3 s_2)$

$c_6 = (\beta_4 l_1 + \beta_3 l_1 s_3 + \beta_4 l_2 s_3 + \beta_4 l_3 s_2)$

$c_7 = \beta_4 l_1 s_3$

$b_{m0} = \beta_4 l_3$

$a_{m4} = 1$

$a_{m3} = (\beta_{m1} + l_2 + s_3), a_{m2} = (\beta_{m2} + l_1 + \beta_{m1} l_2 + \beta_{m1} s_3 + l_2 s_3 + l_3 s_2)$

$a_{m1} = (\beta_{m3} l_3 + \beta_{m2} s_3 + l_1 s_3 + \beta_{m1} l_2 s_3 + \beta_{m1} l_3 s_2)$

$a_{m0} = \beta_{m4} l_3$

$b_{l2} = \beta_{l4}, b_{l1} = (\beta_{l1} \beta_{l4} + \beta_{l4} l_2), b_{l0} = (\beta_{l2} \beta_{l4} + \beta_{l4} l_1 + \beta_{l1} \beta_{l4} l_2)$

$a_{l3} = (\beta_{l1} + \beta_{l3} + l_2 + s_3), a_{l2} = (\beta_{l2} + \beta_{l4} + l_1 + \beta_{l1} \beta_{l3} + \beta_{l1} l_2 + \beta_{l3} l_2 + \beta_{l1} s_3 + l_2 s_3 + l_3 s_2)$

$a_{l1} = (\beta_{l1} \beta_{l4} + \beta_{l2} \beta_{l3} + \beta_{l3} l_1 + \beta_{l4} l_2 + \beta_{l2} s_3 + l_1 s_3 + \beta_{l1} \beta_{l3} l_2 + \beta_{l1} l_2 s_3 + \beta_{l1} l_3 s_2)$

$a_{l0} = (\beta_{l2} \beta_{l4} + \beta_{l4} l_1 + \beta_{l1} \beta_{l4} l_2)$

$$P = \begin{bmatrix} \frac{2\beta_{n1}}{\alpha+1} + \beta_{n2}^2 & l_1\beta_{n2} & l_2\beta_{n2} & 0 & -l_3\beta_{n2} & 0 \\ l_1\beta_{n2} & l_1 & l_1 l_2 & 0 & -l_1 l_3 & 0 \\ l_2\beta_{n2} & l_1 l_2 & l_2^2 + s_2^2 + 1 & s_2\beta_{n3} & s_2 s_3 - l_2 l_3 & -s_2 \\ 0 & 0 & s_2\beta_{n3} & \frac{2\beta_{n3}}{\alpha+1} + \beta_{n3}^2 & s_3\beta_{n3} & -\beta_{n3} \\ -l_3\beta_{n2} & -l_1 l_3 & s_2 s_3 - l_2 l_3 & s_3\beta_{n3} & l_3^2 + s_3^2 & -s_3 \\ 0 & 0 & -s_2 & -\beta_{n3} & -s_3 & \beta_{n4}^2 + 1 \end{bmatrix}$$

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
