# Peer review of "Finite-Time Backstepping Control for Electro-Hydraulic Servo System via Extended State Observer with Perturbation Estimation Performance Improvement"

_machines, doi:10.3390/machines10121163_

Round 1
Reviewer 1 Report
Introduction:
Some imprecise expressions are used in the jargon of control theory, specifically in the use of the word "disturbance" (as in line 36) and "control quantity" (line 14). Disturbances are usually referred to as those external signal sources such as sensor noise that affect the controlled variables and can not be avoided, while the uncertainty originates in errors in the construction of the dynamic model that represents the system either parametric, structural, or unmodeled dynamics. Both sources constitute the system's perturbations (a more precise term than "disturbance" in some parts of the introduction section). To deal with disturbances there are many alternatives ranging from trying to estimate them in the model, to considering them as high-frequency noise that can be filtered out, among other approaches, while robust control strategies (some such as plant augmentation with uncertainty models were not mentioned in the introduction section) deal with parametric or structural uncertainty, in addition to the adaptive approach mentioned in the introduction. The introduction does not show precisely what the research gap is, it shows background information about control approaches and their limitations in dealing with disturbances and parametric uncertainty, without stating exactly what is the specific problem addressed in this work. In the abstract, they refer (confusingly) to reducing the effect of the controlled variable on the transfer function from the disturbance to the disturbance estimation.
However, the contribution reported in the introduction section is declared as an improved estimation speed of an observer and the improved speed of finite-time convergence and tracking accuracy of a backstepping algorithm.
I recommend looking for coherence between the abstract and introduction and stating precisely the gap, the problem, and the solution in this section.
Section 2.1 Cite references for the dynamic model of equations (1) to (3). What is the justification for modeling the load dynamics as linear? idem for the linear equation relating load flow with load pressure? could be stem position a better name for spool position (line 144)?
Section 4:
The affirmation in line 355 "converges to equilibrium point in finite time despite external disturbance and system uncertainty" needs some support. The only source of perturbation taken into account by the dynamic model is the lumped uncertainty, Delta, in equation (5) counting for both, external disturbance on load pressure and unmodeled dynamics of the actuator. What kind of physical "external disturbance" could be managed by the disturbance estimator?
Section 5.2
Experimental results in figure 4 show that the total disturbance deals with uncertainty and unmodeled dynamics of the hydraulic actuator and that the linear assumption about the dynamics of the load is valid. However, the controlled variable needs to deviate from the reference to the estimator to achieve uncertainty compensation. How the experiments could show the performance of the proposed strategy due to external disturbances? Why the lumped uncertainty of the load was not considered in the model?
Conclusion:
The contribution is not clear from the experimental results because the research gap is not clear. The results discussion remark on the precision of the control and the finite time convergence of the estimator. However, this system was solved before using other approaches (for example Asymptotic output tracking control of electrohydraulic systems with unmatched disturbances Guichao Yang, Jianyong Yao, Guigao Le, Dawei Ma, IET Control Theory & Applications), why these remarked improvements of this work are substantially better than other results? Were the precision and time convergence of other contributions to this kind of control problem not good enough?
Some possible minor writing errors:
line 380: a step signal?
Author Response
We (the authors) would like to express our sincere appreciation to the Reviewer for his/her valuable comments and suggestions and his/her time and efforts spent helping us improve the paper. Following the comments made, the paper has been duly revised. A point-wise reply follows after in the attachment.
It is worth pointing out that the Reviewer's comments and suggestions have constructively helped us further improve this manuscript's presentation and quality. For further and more specific changes made, please refer to the detailed responses attached (below) and also the duly revised manuscript. All the changes are marked in blue colour in the revised manuscript.
(i) According to the Reviewer's comments, we corrected some imprecise expressions in the jargon of control theory.
(ii) According to the Reviewer's comments, we rewrote the Introduction section to bring more coherence between the Abstract and Introduction and highlight the issues addressed.
(iii) According to the Reviewer's comments, we rebuilt the test bed and redid the experiment to highlight the research content.
(iv) According to the Reviewer's comments, we changed some places where there were errors in the expression.
Finally, thank you very much for your continued time and effort in handling our paper. We trust that the new version of our work has suitably addressed your concerns.

Reviewer 2 Report
The content of the article titled: "Finite-time Backstepping Control for Electro-hydraulic Servo System via Extended State Observer with Disturbance Estima-tion Performance Improvement" successfully researched finite-time step-back control for Electro-hydraulic Servo Systems force. The results obtained are very attractive, but cannot be published at the present time, need to be edited and improved by the author according to the following suggestions:
-Check the abbreviations in the manuscript and fully add titles and labels for the acronyms in all parts of the manuscript.
- The author needs to edit the references according to the standard form of the journal.
- The author should clearly state the name of the software that the author used for the study, the library package used, the initial parameters of the model, the results compared, and the device name, manufacturer, and accuracy of test equipment, etc.
-For analytical formulas that need the author to cite the source to build, the author needs to add a reference to the related formulas, if it is a new formula built in analytical form, it should be added to the software used to do that.
- The author needs to rearrange the structure of the scientific manuscript content according to the structure of 4 parts: introduction, research methods, results & discussion, and conclusion. In the Research Methods section which includes sections 2, 3 and 4 of the manuscript, the results and discussion are in section 5 and finally the conclusion.
- It is necessary to add the results obtained previously and present the new results obtained in the introduction and conclusion to highlight the content of the manuscript. Besides, it is possible to change the title of the article content to suit the content. Finite-time step-back control for electro-hydraulic Servo Systems via Extended Status Observer with Improved Noise Estimation Performance
- The author needs to explain why he chose the extended state observer to study and control the finite time step back for the Electro-Hydraulic Servo System. With this theoretical method, what advantages and disadvantages do we have in stress research, and need references to prove the author's statement?
-The results obtained on the finite-time step-back control for the electro-hydraulic Servo System by simulation method, compared with the experimental results, should be added directly by the author on the results. Simulation results are obtained to clarify the used formulas. In addition, the author needs to answer the following questions:
+ Why are the noise estimates of the traditional linear ESO affected by the change in the control quantity
+ Why with the new frequency value can remove the control variables in the transfer and reconduction functions and confirm the dependence of the high frequency domain noise range estimation and the observer, the author needs to clearly explain this problem.
+ These results need to be compared, compared with the results obtained previously to verify the accuracy of the results obtained.
-All of the above requirements need to be fully explained by the author and added to the manuscript content to increase the attractiveness and highlight the outstanding issues.
Congratulations to the author on their success with this useful work.
Author Response
First of all, we (the authors) would like to express our sincere gratitude to the Reviewer for their time, effort and recognition given to our manuscript. Your suggestions are very important to improve the quality of this paper.
Thanks a lot and best regards. It is worth pointing out that the Reviewer's comments and suggestions have constructively helped us further improve this manuscript's presentation and quality. The reviewers' encouragement and affirmation of this manuscript gave us great support. For further and more specific changes made, please refer to the detailed responses in the attachment (below) and also the duly revised manuscript. All the changes are marked in blue colour in the revised manuscript.
(i) According to the Reviewer's comments, we checked abbreviations and references and made changes where inappropriate
(ii) According to the Reviewer's comments, to make the expression of the experimental part clearer, we filled in the name, parameters and manufacturer of the equipment in detail and supplemented the name of the software used for research, the library package used, and the initial parameters of the model when conducting experiments
(iii) According to the Reviewer's comments, we add references to the related formulas.
(iv) According to the Reviewer's comments, we adjusted the manuscript's structure.
(v) According to the Reviewer's comments, we rewrote the Introduction section to bring more coherence between the Abstract and Introduction and highlight the issues addressed.
(vi) According to the Reviewer's comments, we rebuilt the test bed and redid the experiment to highlight the research content.
Finally, thank you very much for your continued time and effort in handling our paper. We trust that the new version of our work has suitably addressed your concerns.

Round 2
Reviewer 2 Report
- The authors have edited and supplemented the content and have met the editor's requirements. Finished to publish online, it is recommended that the authors check the entire English style and grammar and revise the content of the final manuscript into 4 parts: Part 1 Introduction (part 1 of the manuscript), Part 2 Research methods (parts 2, 3 of the manuscript), Part 3 Results and discussion (part 4 of the manuscript), Part 4 Conclusions (section 5 of the manuscript).
- After the authors complete the above work, they will be accepted for publication without having to send it back to the reviewer.
Congratulations on the success of the authors
Author Response
I am delighted that you are interested in this paper, and thank you for your approval and suggestions. According to your suggestion, the paper has made the following revisions again, and you can see the details in the updated manuscript through the "Track Changes" function.
(i) Based on the reviewers' suggestion, we restructured the paper from five parts to four.
(ii) Based on the reviewer's comments, we revisited the entire manuscript and changed the sentences that had grammatical issues.
Finally, thank you very much for your continued time and effort in handling our paper.